# Learning from Others: Similarity-based Regularization for Mitigating Artifacts

## Abstract

Common methods for mitigating spurious correlations in natural language understanding (NLU) usually operate in the output space, encouraging a main model to behave differently from a bias model by down-weighing examples where the bias model is confident. While improving out of distribution (OOD) performance, it was recently observed that the internal representations of the presumably debiased models are actually more, rather than less biased. We propose SimgReg, a new method for debiasing internal model components via similarity-based regularization, in representation space: We encourage the model to learn representations that are either similar to an unbiased model or different from a biased model. We experiment with three NLU tasks and different kinds of biases. We find that SimReg improves OOD performance, with little in-distribution degradation. Moreover, the representations learned by SimReg are less biased than in other methods.

## 1 Introduction

Recent studies (McCoy et al., 2019; Geirhos et al., 2020, *inter alia*) show that in many cases neural models tend to exploit spurious correlations (a.k.a dataset biases, artifacts) in datasets and learn shortcut solutions rather than the intended function. For example, in MNLI—a popular Natural Language Understanding dataset—there is a high correlation between negation words such as "not, don't" and the contradiction label (Gururangan et al., 2018). Thus models trained on MNLI confidently predict contradiction whenever there is a negation word in the input without considering the whole meaning of the sentence. As a result of relying on such shortcuts, models fail to generalize and perform poorly when tested on out-of-distribution data (OOD) in which such associative patterns are not present (McCoy et al., 2019) — these models are commonly known as 'biased' models —. Moreover, this behavior limits their practical applicability in cases where the real-world data distribution differs from the training distribution.

Recent efforts to mitigate learning spurious correlations (a.k.a debiasing methods) downweigh the importance of training samples that contain such correlations, effectively performing data reweighting Schuster et al. (2019); Utama et al. (2020a); Sanh et al. (2021); Cadene et al. (2019). Typically, a bias-only model is trained and its confidence is used to reweigh training samples. One might expect that such an *extrinsic* debiasing would lead to "suppressing the model from capturing non-robust features" Du et al. (2022). However, Mendelson & Belinkov (2021) showed a counter-intuitive trend: the more extrinsically de-biased a model is, the more biased are its representations. i.e., higher accuracy of such models on OOD challenge sets is correlated with increase of intrinsic bias. [1] Such superficial debiasing is problematic as the bias may reappear when the model is used in another setting (Orgad et al., 2022).

Inspired by this finding, we propose to perform *intrinsic* debiasing on the internal model components. We develop SimReg, a new debiasing method based on similarity-regularization, where we encourage the internal representations to be either (i) similar to representations of an *unbiased model* [2]; or (ii) dissimilar from representations of a biased model. We further apply the (dis)similarity regularization on either the model representations or its gradients.

---

[1] we measure representation-bias (intrinsic bias) by the *easiness* of identifying the spurious correlations in the representations (more details in Sec 6.1)

[2] In our case, unbiased model is a model that does not rely on the spurious correlations in its decision making.

Our regularization framework allows us to encourage certain constraints on how the data should be represented (representation regularization), or how the model should be sensitive to the representations of the data (gradient regularization). This is different from previous methods, where the main model usually learns to avoid the errors of bias models. Using our approach allows us to transfer knowledge from other models regarding "good" representation of the data and encourage avoiding "biased" representations.

We evaluate our approach on three tasks—natural language inference, fact checking, and paraphrase identification—and multiple spurious correlations attested in the literature: lexical overlap, partial inputs, and unknown biases from weak models (see Section 2.1). We demonstrate that our approach improves performance on out-of-distribution (OOD) challenge sets, while incurring little degradation in in-distribution (ID) performance. Finally, by measuring bias extractability, we find that SimReg representations are less biased than those obtained with competing debiasing methods.

## 2 RELATED WORK

A growing body of work has revealed that models tend to exploit spurious correlations found in their training data (Geirhos et al., 2020). Spurious correlations are correlations between certain features of the input and certain labels, which are not causal. Models tend to fail when tested on out of distribution data, where said correlations do not hold. We briefly mention several relevant cases and refer to Du et al. (2022) for a recent overview of shortcut learning and its mitigation in natural language understanding.

### 2.1 DATASET BIAS

**Partial-input bias.** A common spurious correlation in sentence-pair classification tasks, like natural language inference (NLI), is **partial-input bias** – the association between words in one of the sentences and certain labels. For example, negation words are correlated with a 'contradiction' label when present in the hypothesis in NLI datasets (Gururangan et al., 2018; Poliak et al., 2018) and with a 'refutes' label when present in the claim in fact verification datasets (Schuster et al., 2019). A common approach for revealing the presence of such spurious correlations is to train a partial-input baseline (Feng et al., 2019). When such a model performs well despite having access only to a part of the input, it indicates that that part has spurious correlations.

**Lexical overlap bias.** Another common bias is when certain labels are associated with **lexical overlap** between the two input sentences. McCoy et al. (2019) found that high lexical-overlap between the premise and hypothesis correlates with 'entailment' in NLI datasets. As a result, NLI models fail when evaluated on HANS, a challenge set where that correlation does not hold. Similarly, Zhang et al. (2019) found that models trained on a paraphrase identification dataset fail to predict 'non-duplicate' questions that have high lexical-overlap.

**Unknown biases.** Identifying the preceding biases assumes prior knowledge of the type of bias existing in the dataset. A few studies have used weak learners to identify **unknown biases** (Sanh et al., 2021; Utama et al., 2020b). When limiting either the model capacity or its training data, it tends to exploit simple patterns.

### 2.2 DEBIASING METHODS

Spurious correlation mitigation can be performed on different levels: Data-based mitigation, where the data is augmented with samples that do not align with the bias found in the dataset (Wang & Culotta, 2021; Kaushik et al., 2020, inter alia). Model/training-based mitigation, where the either the model or the training procedure is modified. A common strategy in this approach is to train a *bias model*, which latches on the bias in the dataset, and use its outputs to train the final, debiased, *main model*. He et al. (2019) and Clark et al. (2019) used variants of product-of-experts (PoE) to combine the outputs of the biased and main model during training to encourage the main model to "ignore" biased samples. Utama et al. (2020a) proposed *confidence regularization*, where they perform self-distillation with re-weighted teacher outputs using bias-weighted scaling, i.e., they induce the model to be less confident on biased samples. These methods can be viewed as data re-weighting methods, similar to Liu et al. (2021), who proposed to up-weigh examples that are miss-classified by the

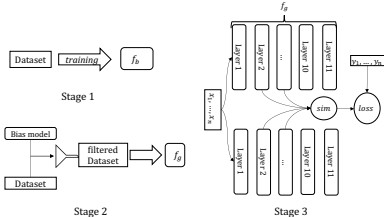

Figure 1: Illustration of SimReg: (1) train a bias model $f_b$; (2) use its predictions to filter the training set and train a target model $f_g$; (3) train a main model while regularizing its representations to be similar to $f_g$.

biased model, i.e., hard examples. All these methods work in the output space, while we work in representation space.

Most relevant to our work, Bahng et al. (2020) debias vision models by learning representations that are statistically independent from those of a biased model, by minimizing a statistical independence measure (HSIC) in a min-max optimization objective. We propose a simpler objective function, based on similarity regulariation, which can easily be trained by SGD. Additionally, while they focus only on learning representations independent of a biased model, we propose learning representations that are either dissimilar from biased models or similar to unbiased ones.

## 2.3 KNOWLEDGE DISTILLATION

Our approach have some similarity with Knowledge-Distillation (KD) methods, where we transfer knowledge from teacher model to a student model (usually to a smaller model to 'compress' the teacher model). In our framework we utilize this transfer of knowledge to improve robustness of a model. Aguilar et al. (2020) perform KD using internal representations, by minimizing the cosine similarity between the representations of the two models. They compare the similarity of the classification token *(CLS)* whereas we compare all of the tokens. Additionally we use second-order isomorphism methods to compare the models whereas they perform first-order methods.

## 3 METHODOLOGY

The key idea behind our approach is to encourage our model to be either similar to an unbiased model or dissimilar from a biased model. To achieve this, we design a three-stage procedure (Figure 1):

1. We train a bias model, $f_b$, on the original training set, $\mathcal{D}$. This model is meant to capture dataset biases, as explained in Section 3.1. In the case of decreasing similarity, we use $f_b$ as our target model, $f_g$, and continue directly to Stage 3.

2. In order to obtain an unbiased model, we filter the training set based on the predictions of $f_b$ and train a target model $f_g$ on the unbiased part of the training set, $\mathcal{D}^{\mathcal{U}}$ (Section 3.2).

3. We train the main model on $\mathcal{D}$ while encouraging its representations to be (dis)similar to those of $f_g$ (Section 3.3).

## 3.1 TRAINING A BIASED MODEL

To mitigate a specific bias, we use a bias-specific model, $f_b$, which is designed to capture the intended bias. To mitigate lexical-overlap bias, we use the model proposed in (Clark et al., 2019): an MLP whose input features are the ratio of overlap between the two parts of the input, and the average of the minimum cosine similarity between the embeddings of each word from the two sentences. For partial-input bias, we use a partial-input model (Gururangan et al., 2018; Belinkov et al., 2019), i.e., we train a model only on the hypothesis/claim part of the input for MNLI/FEVER partial-input biases, respectively. To mitigate unknown biases, we use TinyBert (Turc et al., 2020) as a our bias model ($f_b$);Sanh et al. (2021) showed that limited capacity models recover previously-known biases in the dataset without explicitly modeling them.

In the case of decreasing dissimilarity from a biased model, we use this $f_b$ as the target model, i.e., $f_g = f_b$, and proceed to Stage 3 (Section 3.3). In the case of increasing similarity to an unbiased model, we cannot use $f_b$ as we need an unbiased model; the next section describes how to obtain it.

## 3.2 Obtaining an unbiased model

To obtain an unbiased model, we run $f_b$ on the training set, $\mathcal{D}$, and exclude samples on which $f_b$ is correct and confident. The remaining samples compose our unbiased dataset, $\mathcal{D}^{\mathcal{U}}$:

$$\mathcal{D}^{\mathcal{B}} = \{x_i | x_i \in \mathcal{D} \wedge f_b(x) == y_i \ \wedge \ c(f_b(x_i)) > c_t\} \tag{1}$$

$$\mathcal{D}^{\mathcal{U}} = \mathcal{D} \setminus \mathcal{D}^{\mathcal{B}} \tag{2}$$

where $c_t$ is a confidence threshold and $c(\cdot)$ is the models' confidence, i.e., the highest probability it assigns among the predictions. Our unbiased model, $f_g$, is obtained by training a new model on $\mathcal{D}^{\mathcal{U}}$.

Choosing the threshold $c_t$ is performed manually by plotting the confidence of the bias model over the training set. When there is a significant bias signal in the dataset, we see a spike in the number of biased samples. Figure 2 shows an example for claim-only bias in FEVER.

A natural question is the following: *What is the advantage of our framework if we already have an unbiased model?* We emphasize that the unbiased model was trained on $\mathcal{D}^{\mathcal{U}}$, a subset of $\mathcal{D}$, and argue that other samples in $\mathcal{D}$ could also be useful. Indeed, we show experimentally that training a model on the full training set while regularizing it to be similar to the unbiased model leads to a better IID–OOD tradeoff.

## 3.3 Training the main model

The final step is to train the main model, $f_m$. We propose two approaches. The first is to encourage the model during training to learn different representations than a biased model, by penalizing its similarity to said biased model, $f_b$ (in this case, $f_g = f_b$). Thus the model would learn different decision boundaries than the biased model. The second approach is to increase the similarity of the learned representations to an unbiased model, $f_g$. Thus, our model will encode the data in an unbiased manner and its predictions will be less biased.

In both cases, we need to compute the the similarity between the representations of the main model and those of the target model, $f_g$. Directly comparing the representations of the models on a single example is not possible, since each model might learn a different latent space for representing the data. Furthermore, the two models might have different architectures and dimensionalities. For instance, in some of our experiments we compare BERT-base (768 dimensions) with TinyBERT (128 dimensions) or with an MLP of 7 dimensions. To overcome these challenges, we use second-order similarity measures, which operate at the batch level (Section 3.3.1).

Formally, we add a similarity regularization term to the batch training loss to promote the similarity/dis-similarity. Given a batch $\mathcal{B}$, we minimize the following objective:

$$\mathcal{L} = \sum_{i \in \mathcal{B}} \mathcal{L}_{CE}(f_m(x_i), y_i) + \lambda \cdot sim(Z, H) \tag{3}$$

where $\mathcal{L}_{CE}$ is the cross-entropy loss, $\lambda$ is a trade-off hyper-parameter, $Z$ and $H$ are respectively the main and target model representations of the batch, $f(x)$ is the prediction of the model on input $x$, and $sim$ is a similarity measure. To increase the similarity, we use $\lambda < 0$. Unless otherwise noted, we set $\lambda = 10 \ (-10)$ for decreasing (increasing) similarity.

Since we wish the main model, $f_m$, to resemble or differ from $f_g$ only on biased samples, we apply regularization only on the biased subset, $\mathcal{D}^{\mathcal{B}}$: We stochastically sample a batch either from $\mathcal{D}^{\mathcal{U}}$ and optimize regular cross-entropy, or from $\mathcal{D}^{\mathcal{B}}$ minimizing the objective in Eq. 3. Appendix A.4) shows that regularizing only $\mathcal{D}^{\mathcal{B}}$ results in better OOD performance, supporting our intuition.

Finally, we examine two techniques to increase similarity/dis-similarity of the main and target models. The first works directly at the representations level, i.e., we regularize the similarity of the representations (activations). The second approach is indirect: We regularize the similarity of the *gradients* of the main and target models, inducing similar (different) changes to model weights,

which should make the model more similar (different) to an unbiased (biased) model. In this case, we replace the second term in Eq. 3 with $sim(\nabla Z, \nabla H)$.

### 3.3.1 CHOICE OF SIMILARITY

Instead of directly comparing two vectors that originate from different models, we use second-order isomorphism methods. In particular, as in representational similarity analysis (RSA) Kriegeskorte et al. (2008), we compute a representation similarity matrix (RSM) for each model, which is a matrix where each entry is the similarity between two representations in the batch: $RSM(Z) = \hat{Z}^T \hat{Z}$, where $\hat{Z}$ is column-wise $L_2$ normalization. Then, we compare two RSMs via their correlation:

$$CosCor(Z, H) = \left| \frac{(RSM(Z) - \overline{RSM(Z)}) \cdot (RSM(H) - \overline{RSM(H)})}{|RSM(Z)|_f |RSM(H)|_f} \right| \tag{4}$$

where $\overline{Z}$ is the mean of Z entries and $|Z|_f$ is the Frobenius norm.

As an alternative, we consider a popular RSA-based measure called centered kernel alignment (CKA; Kornblith et al. 2019). CKA is computed in a similar manner to CosCor, replacing the cosine distance with dot-product in the RSM computation and using a different centring transformation.

Experimentally, we found that $CosCor$ works better, so we report its results in the main paper, and compare with CKA in Appendix A.2.

## 4 EXPERIMENTAL SETUP

### 4.1 DATASETS

#### 4.1.1 NATURAL LANGUAGE INFERENCE

We train models on MNLI, a popular NLI dataset consisting of $\sim 400k$ English examples in multiple genres (Williams et al., 2018). Each example is a pair of premise and hypothesis sentences, and the task is to predict whether the hypothesis is entailed, contradicted, or neutral w.r.t the premise. MNLI contains several spurious correlations as discussed in Section 2, such as lexical overlap and hypothesis-only biases. We train on the MNLI training set and report IID results on dev-matched.

As OOD test set, we use HANS (McCoy et al., 2019) for evaluation against **lexical overlap bias**. HANS is constructed using structured templates that obey bias heuristics, e.g., the hypothesis overlaps with premise, but with half of the examples having non-entailment labels, as opposed to the bias in MNLI. For **hypothesis-only bias** we use MNLI-hard, a subset of MNLI's dev-mismatched set where a hypothesis-only model failed to classify correctly (Gururangan et al., 2018).

#### 4.1.2 SYNTHETIC MNLI

As a sanity test, we introduce synthetic spurious correlations to MNLI (Synthetic-MNLI), following prior work (He et al., 2019; Sanh et al., 2021; Dranker et al., 2021). We prepend the input with a 'label-token' that correlates highly with the label. We used tokens *<0>*, *<1>*, and *<2>*, corresponding to *entailment*, *neutral*, and *contradiction*. Following (Dranker et al., 2021), we denote the probability of injecting a token to the input as *prevalence* of the bias, and the probability of the prepended token being correct as the *strength* of the bias. Through all our experiments, we used prevalence of 1.0 and strength of 0.95. The subsets of examples containing bias token with wrong and right correlations are denoted *anti_bias* and *bias* subsets, respectively.

The goal of this setting is to demonstrate the viability of the proposed approaches. Thus we use an *oracle unbiased* model as $f_g$ for the case of increasing similarity, i.e., a model trained on regular MNLI (without synthetic bias). For the bias model, $f_b$, we train a model for a small enough number of steps to capture the bias, judging by the rapid drop of training loss; we found 1k steps sufficient.

#### 4.1.3 FACT VERIFICATION

Fact Extraction and VERification (FEVER) (Thorne et al., 2018) is a dataset for fact verification against textual sources. Given evidence and claim sentences, the task is to predict the relation be-

tween them: SUPPORTED, REFUTED, or NOT ENOUGH INFO. We train on the FEVER training set and evaluate IID on the development set.

We use FEVER-Symmetric Schuster et al. (2019) for OOD evaluation against **claim-only bias**. The construction of FEVER-Symmetric ensures that there is no correlation between partial input and labels, thus it enables us to evaluate the extent of debiasing on this type of bias.

### 4.1.4 QQP

Quora Question Pairs (QQP) is a collection of >400K question pairs from the Quora platform. Given a pair of questions, the task is to predict whether they are *duplicate* (paraphrase) or *non-duplicate*. QQP is biased in that question pairs with low **lexical-overlap** between them are correlated with the *non-duplicate* label. We train on the QQP training set and evaluate IID on the development set.

Paraphrase Adversaries from Word Scrambling (PAWS) Zhang et al. (2019) is a dataset for paraphrase identification that is built in a adversarial manner to lexical-overlap bias. The authors scramble the words of a sentence to generate samples with high lexical-overlap that are not a paraphrase. We use the QQP subset of PAWS as our OOD evaluation set for **lexical-overlap bias**. For IID, we report F1-score of 'duplicate' label, and for OOD we report F1 score of 'non-duplicate' label.

## 4.2 Models

We evaluate our approach using BERT (Devlin et al., 2018) as the main and target model. We repeat some of the experiments using DeBERTa (He et al., 2021) to verify that our method is not specific to BERT. For full training details see Appendix A.1.

## 5 Results

### 5.1 Synthetic bias

The results on Synthetic-MNLI are in Table 1. All of the SimReg approaches resulted in a large increase compared to the baseline on the *anti-biased* subset, where the synthetic token is mis-aligned with the label. Increasing similarity ($\uparrow$) performed better than decreasing it ($\downarrow$). The improvement comes at a cost of a small decrease on the biased subset, with the exception of increasing gradient

Table 1: Results on Synthetic-MNLI.

| Model | Biased | Anti-biased |
|---|---|---|
| BERT-base | $98.5 \pm 0.1$ | $41.8 \pm 1.1$ |
| Oracle | $83.8$ | $82.1$ |
| SimReg $\uparrow$ | $96.7 \pm 0.1$ | $61.00 \pm 0.9$ |
| SimReg $\downarrow$ | $97.0 \pm 0.0$ | $49.0 \pm 2.4$ |
| $\nabla$ SimReg $\uparrow$ | $82.8 \pm 0.0$ | $\mathbf{72.1 \pm 0.0}$ |
| $\nabla$ SimReg $\downarrow$ | $94.7 \pm 0.4$ | $57.1 \pm 2.4$ |

similarity ($\nabla$ SimReg $\uparrow$), where the drop is large. Compared to an oracle model, which was trained without the synthetic bias, the regularized models perform worse, indicating that they were not able to completely discard the bias.

### 5.2 Known bias

Tables 2 and 3 show the results on known bias cases. Focusing first on partial-input bias (Table 2), all our SimReg models outperform the baseline on the OOD sets (MNLI hard and FEVER Symmetric), except for $\nabla$ SimReg $\downarrow$, which is on par. Increasing similarity ($\uparrow$) seems to work better than decreasing similarity ($\downarrow$). The improvements are comparable to or better than those of competitive approaches (POE and ConfReg).

Turning to lexical-overlap bias (Table 3), we see a similar pattern: SimReg performs much better than the baseline on HANS non-entailed and PAWS (the OOD sets), with little or no degradation on the corresponding IID dev sets. In this bias type, increasing similarity works much better than decreasing it. Compared to competitive methods, ConfReg performs a bit better on HANS non-entailed. However, SimReg $\uparrow$ is much better on average on HANS.

A telling comparison is between SimReg and the guidance model, which is a model that was trained only on unbiased examples ($f_g$, Section 3.2). In almost all cases, when we increase similarity to this model (rows with $\uparrow$), we get models that perform better than it, on both IID and OOD sets. Interestingly, in QQP $f_g$ performed poorly on PAWS, but using it as a guidance model still improved

Table 2: Known bias: Mitigating partial-input bias.

| | MNLI | | FEVER | |
| | dev | hard | dev | Symmetric |
|---|---|---|---|---|
| BERT-base | $83.9 \pm 0.1$ | $76.9 \pm 0.2$ | $\mathbf{85.4 \pm 0.1}$ | $58.2 \pm 0.6$ |
| Guidance | 79.2 | 78.0 | 70.6 | 59.7 |
| POE | $84.1 \pm 0.1$ | $78.02 \pm 0.4$ | $81.44 \pm 0.1$ | $59.2 \pm 0.1$ |
| ConfReg | $84.3 \pm 0.1$ | $\mathbf{78.4 \pm 0.6}$ | $85.2 \pm 0.3$ | $61.0 \pm 1.7$ |
| SimReg ↑ | $\mathbf{84.4 \pm 0.2}$ | $78.2 \pm 0.1$ | $84.5 \pm 0.2$ | $\mathbf{61.2 \pm 0.4}$ |
| SimReg ↓ | $83.0 \pm 0.1$ | $77.9 \pm 0.5$ | $84.1 \pm 0.9$ | $60.3 \pm 1.1$ |
| ∇ SimReg ↑ | $83.1 \pm 0.1$ | $78.2 \pm 0.1$ | $80.9 \pm 0.1$ | $60.1 \pm 0.6$ |
| ∇ SimReg ↓ | $81.7 \pm 0.2$ | $77.8 \pm 0.2$ | $79.9 \pm 0.6$ | $58.3 \pm 0.1$ |

Table 3: Known bias: Mitigating lexical-overlap bias.

| | MNLI | HANS | | | QQP | |
| | dev | non-entailment | entailment | avg | dev | PAWS |
|---|---|---|---|---|---|---|
| BERT-base | $83.9 \pm 0.1$ | $30.1 \pm 1.3$ | $\mathbf{98.1} \pm 0.7$ | $64.1$ | $\mathbf{88.4} \pm 0.1$ | $28.2 \pm 2.3$ |
| Guidance | 83.0 | 24.3 | 89.2 | 56.7 | 87.9 | 29.3 |
| PoE | $83.9 \pm 0.4$ | $41.8 \pm 5.3$ | $93.3 \pm 3.4$ | $67.6$ | $75.4 \pm 0.3$ | $\mathbf{73.6}* \pm 1.7$ |
| ConfReg | $\mathbf{84.3} \pm 0.2$ | $60.9 \pm 6.6$ | $72.3 \pm 8.5$ | $66.6$ | $85.4 \pm 0.5$ | $28.7 \pm 3.6$ |
| SimReg ↑ | $83.6 \pm 0.1$ | $57.6 \pm 6.0$ | $82.3 \pm 2.6$ | $\mathbf{70.0}$ | $86.9 \pm 0.0$ | $36.9 \pm 1.2$ |
| SimReg ↓ | $83.9 \pm 0.1$ | $40.1 \pm 1.7$ | $89.4 \pm 3.6$ | $64.8$ | $87.7 \pm 0.3$ | $32.6 \pm 1.2$ |
| ∇ SimReg ↑ | $83.4 \pm 0.0$ | $57.9 \pm 1.4$ | $73.0 \pm 2.6$ | $65.5$ | $87.3 \pm 0.1$ | $33.2 \pm 1.7$ |
| ∇ SimReg ↓ | $84.0 \pm 0.1$ | $48.8 \pm 6.4$ | $89.8 \pm 5.5$ | $69.3$ | $87.5 \pm 0.2$ | $31.6 \pm 3.5$ |

performance. These results support our hypothesis that increasing similarity to an unbiased model can lead to better representations than those of the unbiased model itself.

## 5.3 UNKNOWN BIAS

The results of unknown bias mitigation are in Table 4. Debiasing from unknown bias seems to be more challenging than known bias mitigation, with smaller improvements of all debiasing methods compared to the baseline. This is consistent with prior work on unknown biases (Sanh et al., 2021; Utama et al., 2020b). On MNLI hard, there is practically no improvement, but in other cases we find improvements of up to 6–7 points (on HANS and PAWS).

Comparing the different SimReg models, increasing similarity (↑) works better than decreasing it (↓), and regularizing representations is generally better than regularizing gradients, except for PAWS, where increasing gradient similarity works very well. When compared to competitive approaches, SimReg works better on HANS and PAWS, but ConfReg is better on MNLI hard and FEVER Symmetric.

## 5.4 RESULTS WITH STRONGER MODELS

In this section we investigate whether our approach improves the performance of stronger models than BERT. While most work tends to compare with BERT as the baseline, it is important to demonstrate that a new debiasing method is effective also when applied to stronger models.[3] We experiment with DeBERTa-V1 (He et al., 2021) and DeBERTa-V3 (Federer et al., 2019). As Table 5 shows, SimReg still leads to improvements above the strong DeBERTa-V3. Notably, ConfReg does

---

[3]Bowman (2022) made such a claim about *analyzing* stronger models; we believe it is similarly important to work on *robustifying* stronger models.

Table 4: Results for unknown bias mitigation.

| | MNLI | | HANS | FEVER | | QQP | |
| --- | --- | --- | --- | --- | --- | --- | --- |
| | dev | hard | avg | dev | Symm. | dev | PAWS |
| BERT | **83.9** ±0.1 | 76.9±0.2 | 64.1 | 85.4±0.1 | 58.2±0.6 | **88.4** ±0.1 | 28.2 ±2.2 |
| Biased | 64.6±0.1 | 51.4±0.2 | 50.0 | 71.3±0.1 | 45.3±0.4 | 80.6 ±0.1 | 30.4 ±0.2 |
| Guidance | 80.9 | 74.9 | 61.9 | 64.9 | 50.9 | 73.13 | 55.4 |
| POE | 83.1±0.1 | 75.8±0.5 | 65.4 | 84.2±0.1 | 57.7±1.7 | 87.6 ±0.4 | 34.7 ±6.1 |
| ConfReg | 83.4±0.4 | **77.0** ±0.5 | 63.2 | **86.0** ±0.2 | **60.0** ±1.6 | 86.6 ±0.3 | 16.6 ±3.0 |
| BERT + $\mathcal{F}_{bow}$ | 83.0 ±0.4 | 76.6 | 70.4 | 87.1 ±0.2 | 61.0 ±1.4 | - | - |
| SimReg↑ | 82.9±0.1 | 76.1±0.1 | **68.2** | 85.6±0.3 | 59.1±0.2 | 84.4 ±0.1 | 40.6 ±0.6 |
| SimReg↓ | 82.9±0.3 | 75.8±0.6 | 63.5 | 84.1±0.5 | 57.5±2.2 | 86.6 ±0.7 | 32.4 ±4.6 |
| ∇SimReg↑ | 83.4±0.0 | 75.9±0.3 | 64.8 | 81.1±0.0 | 59.1±0.1 | 82.2 ±0.3 | 45.5 ±1.5 |
| ∇SimReg↓ | 82.9±0.3 | 75.3±0.4 | 62.7 | 83.8±1.1 | 57.2±1.5 | 84.0 ±2.0 | 36.4 ±13.8 |

Table 5: Results with DeBERTa-V3.

| | MNLI | | HANS | QQP | |
| --- | --- | --- | --- | --- | --- |
| | dev | hard | non-ent | dev | PAWS |
| baseline | 89.9 ±0.1 | 85.2 ±0.1 | 56.7 ±2.2 | 89.9 ±0.1 | 55.7 ±5.6 |
| ConfReg | 90.1 ±0.1 | 86.0 ±0.1 | 54.5 ±2.0 | 88.8 ±0.1 | 61.1 ±2.0 |
| SimReg ↑ | 89.1 ±0.2 | 85.1 ±0.3 | 66.8 ±0.7 | 86.3 ±0.2 | 67.0 ±2.0 |
| ∇SimReg ↑ | 87.9 ±1.6 | 83.9 ±1.1 | 74.8 ±1.3 | 89.3 ±0.1 | 58.2 ±1.5 |

not improve the baseline in this case, while SimReg does. Similar improvements are obtained with DeBERTa-V1 (Table 13, App. A.8).

# 6 ANALYSIS

## 6.1 PROBING FOR BIAS EXTRACTABILITY

Recall that Mendelson & Belinkov (2021) found that, counter-intuitively, extrinsic debiasing methods, like POE and ConfReg, increase the bias in models' internal representations. Their finding motivated us to propose SimReg, which regularizes the representations themselves. Their discovery was made by measuring the *extractability* of bias from model representations, for which they used compression, a measure from minimum description length probing (Voita & Titov, 2020). We perform a similar analysis of our models. Figure 2 shows the increase in compression of debiased models compared to the baseline, against the increase of performance on OOD challenge set HANS (non-entailed). Consistent with Mendelson & Belinkov (2021), ConfReg leads to a large performance increase on HANS, but suffers from a large increase in bias extractability. In contrast, SimReg is able to improve performance on OOD challenge sets with little increase in bias extractability, supporting our motivation to perform debiasing at the representation level.

## 6.2 BIAS RECOVERY

We demonstrate the importance and effectiveness of debiasing the representations of a model. We retrained the classification layer of the resulted models after debiasing. In Table 6 we present the results on lexical-overlap bias; in App.A.6 we present other configurations. In all approaches there was a drop on HANS non-entailment accuracy. In SimReg↑ we got the highest performance on the OOD challenge set. This indicates that the representations produced by SimReg↑ have the least signal of lexical-overlap spurious correlation.

Table 6: lexical-overlap bias recovery

| | IID | HANS - |
| --- | --- | --- |
| POE | 83.4 ±0.2 | 38.2 ±4.2 |
| ConfReg | **84.8** ±0.1 | 20.5 ±6.2 |
| BERT + $\mathcal{F}_{bow}$ | 83.7 ±0.2 | 42.2 ±1.2 |
| SimReg↑ | 84.3 ±0.3 | **44.2** ±4.4 |

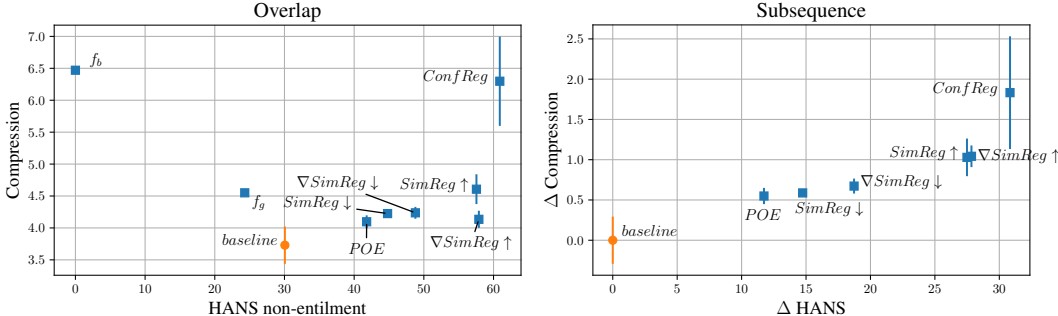

Figure 2: Compression increment relative to $HANS$ non-entailment subset. Left: Overlap bias. Right: Subsequence bias. SimReg leads to lower increase in intrinsic bias than ConfReg.

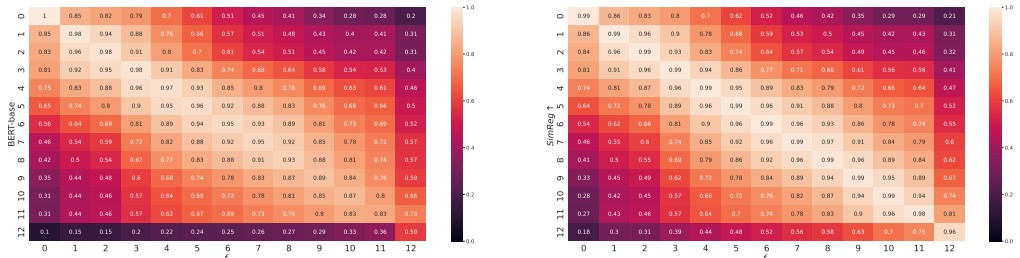

Figure 3: Similarity of an unbiased model, $f_g$, to either a baseline (Left) or a SimReg model (Right). Similarity regularization makes top layers more similar to the unbiased model, as desired.

### 6.3 SIMILARITY HEAT-MAP ANALYSIS

To investigate whether our similarity-based regularization achieves its goal, we compute the similarity between every layer in the main model and every layer in the (unbiased) guidance model, and likewise the similarity between layers of the baseline model and layers of the guidance model. We expect our similarity regularization to increase the similarity of the main model to the guidance model, compared to that of the baseline model.

Figure 3 (Left) shows that, without similarity-based regularization, the bottom layers of the baseline and guidance model are already similar, but the top layers are rather different. This is consistent with findings on how fine-tuning affects mostly the top layers (Mosbach et al., 2020; Merchant et al., 2020), as both models started from a pre-trained BERT. Figure 3 (Right) shows that after our similarity-based regularization, the top layers of the main and guidance models become very similar, as desired. Moreover, the regularization also indirectly affects lower layers (bottom row of the heatmap). We conclude that the similarity regularization is successful and affects large parts of the model even when applied only on a few layers.

### 6.4 ABLATIONS

Our main reported results were with CosCor as the similarity method (Section 3.3.1). In Appendix A.2 we report experiments with CKA as an alternative measure, showing it also leads to consistent improvements compared to the baseline, albeit not as good as CosCor. Additionally, in the main experiments we reported results when regularizing only on batched from a biased subset, $\mathcal{D}^{\mathcal{B}}$. In Appendix A.4, we compare with results when regularizing all samples, finding that regularizing only on the biased samples is better.

## 7 CONCLUSION

In this work, we have proposed SimReg, a new debiasing approach using similarity-based regularization. We have explored several variants of this approach, regularizing by increasing or decreasing similarity of a model to biased or unbiased models, respectively, at the level of either the representations or the gradients. We found SimReg to improve performance on OOD challenge sets on multiple bias types and NLU tasks, with little decrease in IID performance. Future work may investigate the effect of simultaneously learning from unbiased and biased models. Another interesting direction is to extend our approach to generation tasks, which would require different similarity measures.

## ETHICS STATEMENT

Our work develops a new approach to mitigate spurious correlations in NLU tasks. These are also known as dataset biases, but are different from social biases such as gender or racial bias. One could use our approach to debias against social biases. However, a malicious actor could use our basic approach to increase such social bias, rather than decrease it, by reversing the optimization. In such a case, probing for bias extractability as we performed in this work may expose biased representations.

## REPRODUCIBILITY STATEMENT

We provide detailed configurations in Appendix A.1. Code in PyTorch (Paszke et al., 2019) for reproducing our results is available at `https://github.com/simreg/SimReg`. Gradient regularization experiments require twice the memory of representation regularization. For gradients experiments we used an NVIDIA A40 GPU; a typical training time was $\sim 7$ hours. For representations experiments, we used an NVIDIA A4000 RTX GPU, and training time was $\sim 6$ hours.

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

## A    APPENDIX

### A.1    TRAINING PARAMETERS

We used pre-trained *bert-base-uncased* from HuggingFace models (Wolf et al., 2019), trained for 10 epochs with a batch size of 64 and a learning rate of $2\mathrm{e}-5$ that warms up on the first $10\%$ of the training steps and decays linearly. The results reported in the tables are the mean and standard deviation of 3 different random seeds (changes the initialization of the model and the training batches).

We performed the similarity regularization on multiple layers, aggregating the similarities by summing it across the layers. When increasing similarity $f_g$ is the same architecture as $f_m$. Experimentally we found that upper layers are most effective for this case. We regularize layers $10, 11$, and $12$, each layer in $f_g$ paired with its parallel layer in $f_m$. For gradient regularization, we choose layers from across the model: embedding layer and layers 1, 7, 11, and 12, and use this set of layers for increasing and decreasing gradient similarity whenever $f_g$ has the same architecture as $f_m$ (partial-input biases for $\downarrow$).

For decreasing similarity from models with a different architecture, using a combination of layers yielded the best results, and we used the upper two layers and the embeddings layers. For gradient dis-similarity, regularizing the gradient of lower layers was the major factor as validated in Appendix A.3, which provides additional results when regularizing different layers.

### A.2    RESULTS WITH CKA

In Table 7 we repeat experiments from the main paper, but using CKA as the similarity measure instead of CosCor. This approach also leads to improvements compared to the baseline, but not as large as using CosCor.

Table 7: Results of SimReg with CKA as the similarity measure.

|  | MNLI | | HANS | |
| --- | --- | --- | --- | --- |
|  | dev | hard | non-entailment | entailment |
| **Unknown bias** | | | | |
| SimReg $\uparrow$ | $84.1 \pm 0.2$ | $77.1 \pm 0.4$ | $35.5 \pm 1.8$ | $96.4 \pm 0.2$ |
| SimReg $\downarrow$ | $82.4 \pm 0.3$ | $75.0 \pm 0.3$ | $43.7 \pm 3.5$ | $79.0 \pm 2.0$ |
| $\nabla$SimReg $\uparrow$ | $76.5 \pm 0.1$ | $70.7 \pm 0.3$ | $87.9 \pm 0.5$ | $25.5 \pm 1.6$ |
| $\nabla$SimReg $\downarrow$ | $81.8 \pm 0.5$ | $75.2 \pm 0.7$ | $38.8 \pm 3.1$ | $84.4 \pm 3.0$ |
| **Lexical bias** | | | | |
| SimReg $\uparrow$ | $83.6 \pm 0.1$ | $-$ | $57.5 \pm 6.0$ | $82.3 \pm 2.7$ |
| SimReg $\downarrow$ | $83.9 \pm 0.1$ | $-$ | $36.7 \pm 5.8$ | $96.2 \pm 1.4$ |
| $\nabla$SimReg $\uparrow$ | $82.9 \pm 0.2$ | $-$ | $59.0 \pm 4.5$ | $68.7 \pm 5.5$ |
| $\nabla$SimReg $\downarrow$ | $83.8 \pm 0.1$ | $-$ | $41.8 \pm 4.8$ | $93.6 \pm 1.1$ |
| **Hypothesis-only** | | | | |
| SimReg $\uparrow$ | $82.4 \pm 0.2$ | $78.8 \pm 0.2$ | $-$ | $-$ |
| SimReg $\downarrow$ | $82.3 \pm 0.1$ | $76.3 \pm 0.7$ | $-$ | $-$ |
| $\nabla$SimReg $\uparrow$ | $77.0 \pm 0.1$ | $79.1 \pm 0.2$ | $-$ | $-$ |
| $\nabla$SimReg $\downarrow$ | $81.6 \pm 0.3$ | $77.9 \pm 0.7$ | $-$ | $-$ |

### A.3    LAYERS

In the main experiments, we regularized multiple layers together, as described in Appendix A.1. In Tables 8 and 9 we present the results when regularizing different layers, where we regularize only one layer at a time. We see several patterns. In the case of increasing representation similarity, deeper layers work better. In the case of regularizing gradients, early layers work better. In decreasing representation similarity, individual layers are not effective, as opposed to regularizing multiple layers as in the main experiments. To simplify the choice of layers, we reported results with a set of layers as described in Appendix A.1.

Table 8: Layers search.

|  | SimReg↑ | | $\nabla$SimReg ↑ | |
| --- | --- | --- | --- | --- |
|  | bias | anti-bias | bias | anti-bias |
| Embeddings | 98.7 ±0.1 | 39.3 ±0.9 | 96.1 ±0.2 | 63.8 ±0.7 |
| Layer-2 | 98.2 ±0.0 | 45.6 ±1.2 | 96.9 ±0.0 | 59.9 ±0.0 |
| Layer-4 | 98.7 ±1.0 | 38.7 ±12.4 | 97.3 ±0.1 | 56.6 ±0.3 |
| Layer-6 | 97.3 ±0.0 | 56.2 ±0.9 | 98.4 ±1.3 | 42.6 ±20.4 |
| Layer-8 | 96.3 ±0.0 | 59.8 ±1.6 | 98.5 ±1.0 | 45.1 ±11.4 |
| Layer-10 | 94.9 ±0.1 | 68.1 ±0.1 | 98.6 ±0.5 | 41.2 ±9.9 |
| Layer-12 | 96.0 ±0.1 | 62.7 ±1.2 | 99.0 ±0.1 | 35.0 ±2.9 |

Table 9: dis-similarity Layers search.

|  | SimReg↓ | | $\nabla$SimReg ↓ | |
| --- | --- | --- | --- | --- |
|  | bias | anti-bias | bias | anti-bias |
| Embeddings | 98.9 ±0.0 | 35.7 ±0.1 | 94.7 ±0.4 | 57.1 ±2.4 |
| Layer-2 | 98.8 ±0.1 | 37.1 ±0.3 | 77.2 ±14.2 | 40.1 ±34.9 |
| Layer-7 | 99.0 ±0.0 | 38.4 ±0.1 | 91.1 ±10.0 | 43.6 ±21.3 |
| Layer-12 | 98.7 ±0.0 | 36.4 ±1.0 | 100.0 ±0.0 | 0.5 ±0.7 |

## A.4 REGULARIZATION SET

In the main body we have reported results when performing similarity-based regularization only on the biased samples, $\mathcal{D}^{\mathcal{B}}$, as described in Section 3.3. In Table 10 we compare to the case of regularizing on all the training samples. Clearly, it is important to perform regularization only on $\mathcal{D}^{\mathcal{B}}$. This is especially true for the dissimilarity case, where the model regularized on all samples sometimes fails to converge, leading to random performance on MNLI dev and constant predictions on HANS (see last row).

Table 10: Comparing regularizing only on $\mathcal{D}^{\mathcal{B}}$ to regularizing on all samples.

| Bias | | MNLI | HANS | |
| --- | --- | --- | --- | --- |
|  |  | dev | non-entailment | entailment |
| Lexical-overlap | SimReg ↑ | 83.6 ± 0.1 | 57.5 ± 6.0 | 82.3 ± 2.7 |
|  | SimReg↑ (all) | 84.4 ±0.0 | 7.4 ±1.4 | 98.4 ±0.4 |
|  | SimReg ↓ | 83.9 ± 0.1 | 40.1 ± 1.7 | 89.4 ± 3.6 |
|  | SimReg ↓ (all) | 83.8 ± 0.2 | 18.5 ± 4.3 | 96.8 ± 1.3 |
|  | $\nabla$SimReg↑ | 83.3 ±0.4 | 56.4 ±2.8 | 75.6 ±0.8 |
|  | $\nabla$SimReg↑ (all) | 46.1 ±52.8 | 61.7 ±54.1 | 44.4 ±62.7 |
|  | $\nabla$SimReg↓ | 84.0 ±0.1 | 48.8 ±6.4 | 89.8 ±5.5 |
|  | $\nabla$SimReg↓ (all) | 32.8 ±0.2 | 100.0 ±0.0 | 0.0 ±0.0 |
| Unknown | SimReg ↑ | 82.9 ± 0.2 | 54.1 ± 1.1 | 82.4 ± 2.5 |
|  | SimReg ↑ (all) | 84.3 ± 0.1 | 7.3 ± 0.7 | 98.3 ± 0.2 |
|  | SimReg ↓ | 82.9 ± 0.3 | 41.4 ± 2.5 | 85.6 ± 4.3 |
|  | SimReg ↓ (all) | 83.1 ± 0.2 | 3.6 ± 0.4 | 98.9 ± 0.2 |
|  | $\nabla$SimReg↑ | 82.7 ±0.1 | 54.4 ±2.5 | 75.0 ±2.4 |
|  | $\nabla$SimReg↑ (all) | 82.8 ±0.0 | 32.7 ±0.1 | 82.7 ±0.0 |
|  | $\nabla$SimReg ↓ | 82.8 ± 0.2 | 40.0 ± 2.0 | 86.2 ± 6.3 |
|  | $\nabla$SimReg ↓ (all) | 32.3 ± 0.6 | 100.0 ± 0.0 | 0.0 ± 0.0 |

## A.5 SYNTHETIC BIAS

In this section we present more detailed results for synthetic-MNLI. In Table 11 we show wider range of configuration for the case of increasing similarity. Note that $\lambda$ values for resulted in models

with better performance on the anti-biased set. Bert-base is BERT trained on synthetic-MNLI, while BERT (Oracle) is trained on MNLI. In Table 12 we present a partial list of decreasing similarity experiments.

Table 11: Increasing similarity: Synthetic-MNLI evaluation for prevalence=1 and strength=0.95, $\nabla$ denotes regularizing gradients.

|  | Biased | Anti-biased | Unbiased |
| --- | --- | --- | --- |
| Bert-base | $98.5 \pm 0.1$ | $41.8 \pm 1.1$ | $78.0 \pm 0.4$ |
| BERT (Oracle) | $83.5 \pm 0.3$ | $82.0 \pm 0.9$ | $84.0 \pm 0.3$ |
| **CosCor** | | | |
| SimReg $\uparrow (\lambda = 1)$ | $97.5 \pm 0.0$ | $57.0 \pm 0.6$ | $82.6 \pm 0.2$ |
| SimReg $\uparrow (\lambda = 10)$ | $96.9 \pm 0.0$ | $59.3 \pm 0.8$ | $83.0 \pm 0.1$ |
| SimReg $\uparrow (\lambda = 100)$ | $96.8 \pm 0.1$ | $60.3 \pm 0.6$ | $82.7 \pm 0.1$ |
| $\nabla$SimReg $\uparrow (\lambda = 1)$ | $96.8 \pm 0.1$ | $60.8 \pm 0.6$ | $82.7 \pm 0.5$ |
| $\nabla$SimReg $\uparrow (\lambda = 10)$ | $95.5 \pm 0.2$ | $56.7 \pm 2.6$ | $81.1 \pm 1.0$ |
| $\nabla$SimReg $\uparrow (\lambda = 100)$ | $82.8$ | $72.1$ | $79.1$ |
| **Linear-CKA** | | | |
| SimReg $\uparrow (\lambda = 1)$ | $97.3 \pm 0.0$ | $57.7 \pm 0.1$ | $83.3 \pm 0.2$ |
| SimReg $\uparrow (\lambda = 10)$ | $96.8 \pm 0.1$ | $60.2 \pm 0.4$ | $83.6 \pm 0.3$ |
| SimReg $\uparrow (\lambda = 100)$ | $96.7 \pm 0.1$ | $61.0 \pm 0.9$ | $83.2 \pm 0.2$ |
| $\nabla$SimReg $\uparrow (\lambda = 1)$ | $98.1 \pm 0.1$ | $31.3 \pm 1.0$ | $70.4 \pm 1.7$ |
| $\nabla$SimReg $\uparrow (\lambda = 10)$ | $96.7 \pm 0.1$ | $17.5 \pm 0.2$ | $51.3 \pm 0.7$ |
| $\nabla$SimReg $\uparrow (\lambda = 100)$ | $32.8 \pm 0.0$ | $32.2 \pm 0.0$ | $32.7 \pm 0.0$ |

Table 12: Decreasing similarity: Synthetic-MNLI evaluation for prevalence=1 and strength=0.95.

| Model | Biased | Anti-biased | Unbiased |
| --- | --- | --- | --- |
| Bert-base | $98.5 \pm 0.1$ | $41.8 \pm 1.1$ | $78.0 \pm 0.4$ |
| Biased-BERT | $99.9$ | $05.7$ | $64.3$ |
| **CosCor** | | | |
| SimReg$\downarrow (\lambda = 1)$ | $99.6 \pm 0.6$ | $12.2 \pm 17.3$ | $53.4 \pm 25.9$ |
| SimReg $\downarrow (\lambda = 10)$ | $97.0 \pm 0.1$ | $49.0 \pm 2.4$ | $76.1 \pm 2.0$ |
| SimReg$\downarrow (\lambda = 100)$ | $33.5 \pm 1.6$ | $32.3 \pm 0.4$ | $33.3 \pm 1.6$ |
| **Linear-CKA** | | | |
| SimReg $\downarrow (\lambda = 1)$ | $99.2 \pm 0.1$ | $21.9 \pm 6.8$ | $69.0 \pm 0.4$ |
| SimReg $\downarrow (\lambda = 10)$ | $32.3 \pm 0.7$ | $32.2 \pm 0.0$ | $32.3 \pm 0.6$ |
| SimReg $\downarrow (\lambda = 100)$ | $32.3 \pm 0.7$ | $32.6 \pm 0.6$ | $32.3 \pm 0.7$ |

## A.6 BIAS RECOVERY

Here we present the full results of our bias-recovery experiments.

## A.7 THRESHOLD CHOOSING

## A.8 RESULTS WITH DEBERTA-V1

Table 13 provides results with DeBERTa-V1. SimReg consistently improves the baseline, demonstrating its success with stronger models than BERT.

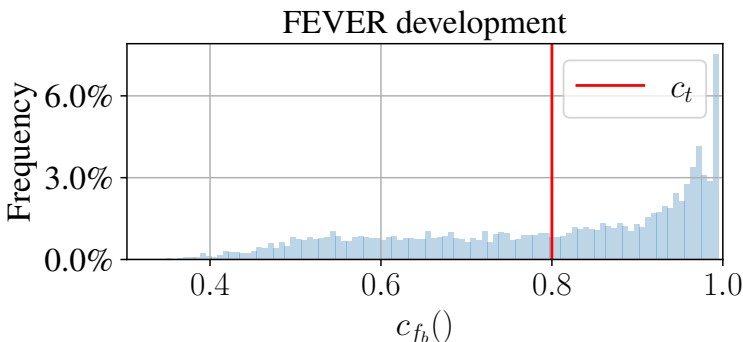

Figure 4: Confidence distribution of a claim-only model ($f_b$) on FEVER; here $c_t = 0.8$.

Table 13: Results with stronger models - DeBERTa V1.

| | Hypothesis | | Lexical | |
|---|---|---|---|---|
| | IID | OOD | IID | OOD |
| baseline | 88.2 ±0.2 | 82.3 ±0.1 | 88.2 ±0.2 | 75.2 ±0.9 |
| **Known bias** | | | | |
| SimReg ↑ | 88.2 ±0.0 | 83.1 ±0.0 | 87.5 ±0.0 | 80.5 ±0.6 |
| SimReg ↓ | 87.0 ±0.6 | 82.1 ±0.3 | 87.8 ±0.1 | 77.3 ±1.5 |
| ∇SimReg ↑ | 87.2 ±0.1 | 82.7 ±0.2 | 87.5 ±0.2 | 79.8 ±0.6 |
| ∇SimReg ↓ | 85.0 ±1.1 | 81.7 ±2.7 | 87.8 ±0.0 | 78.4 ±1.3 |
| **Unknown bias** | | | | |
| SimReg ↑ | 88.3 ±0.1 | 83.0 ±0.3 | 88.3 ±0.1 | 74.6 ±1.5 |
| SimReg ↓ | 87.7 ±0.0 | 82.3 ±0.6 | 87.7 ±0.0 | 76.5 ±0.3 |
| ∇SimReg ↑ | 85.4 ±0.2 | 80.8 ±0.4 | 85.4 ±0.2 | 73.7 ±0.7 |
| ∇SimReg ↓ | 87.3 ±0.1 | 81.8 ±0.4 | 87.3 ±0.1 | 78.5 ±1.6 |

