# OpenReview forum: " Learning from Others: Similarity-based Regularization for Mitigating Artifacts"
_ICLR.cc/2023/Conference — Submitted to ICLR 2023_

### Official Review · Reviewer_UAbR · 2022-10-20

**Confidence:** 4
**Correctness:** 3
**Technical Novelty And Significance:** 3
**Empirical Novelty And Significance:** 3
**Recommendation:** 5

**Clarity, Quality, Novelty And Reproducibility:**

The paper is generally clear. The approach is closely tied to existing methods such as distillation (see above), though I haven't seen it applied for this purpose.

Typos and such:
Section 3.3: the the


**Strength And Weaknesses:**


Strengths:
- A simple approach for mitigating artifacts
- Results for SimReg+ are generally better than most baselines
- The second-order similarity measure seems interesting
- The internal bias representation results are interesting

Weaknesses:
- The proposed approach is somewhat counter-intuitive: why would making a model imitate a weak model (i.e., one trained on less data) result in a stronger model? This is sort of like doing knowledge distillation where the teacher is the smaller model. I see that empirically the idea sort-of works, although sometimes it does much worse (e.g., HANS non-entailment, PAWS) and in some cases the differences are small (Table 2). I wonder how generalizable this approach is.
- Related to the previous point - for some reason the Guidance baseline is missing from the Deberta experiments (Table 12 doesn't show any baselines except the original model).
- Given that SimReg+ works best among all proposed methods, it might make sense to focus on it, and present the rest as ablations

Other questions:
- It is not clear why the authors chose HANS- as the X axis in Figure 3.

**Summary Of The Paper:**

This paper proposes a regularization method for mitigating artifacts. Their method (SimReg+) is simple: take a model that is known to be artifact-free, and train another model to be similar to that model. The idea is somewhat counter-intuitive, but works reasonably well on some datasets, and is also shown to have fewer internal biases. The authors also propose several variants of the main idea (e.g., decreasing the similarity between the model and some model that is known to be biased), but those do not work as well as the main idea.

**Summary Of The Review:**

The paper presents a simple method that seems to overall work well. However, it is not entirely clear how generalizable it is, given that it is not very intuitive and the results are sometimes mixed.

---

> ### Author Response · Authors · 2022-11-17
> **Response to reviewer UAbR**
>
> Thank you for your review, we are glad that you liked our method!
> In addition to the points below, we have added an additional experiment, we hope you find it insightful and improve the paper's quality.
>
> > The proposed approach is somewhat counter-intuitive: why would making a model imitate a weak model (i.e., one trained on less data) result in a stronger model?
>
> please see the first comment.
>
> > for some reason the Guidance baseline is missing from the Deberta experiments
>
> Thanks for your note. We have added it (it was removed to save space), and we are working on adding additional configurations for DeBERTa.
> > “It is not clear why the authors chose HANS- as the X axis in Figure 3.”:
>
> We choose HANS- (non-entailment) following the work by Menselson and Belinkov, which showed that the intrinsic bias (compression) is positively correlated with increase of accuracy on OOD challenge sets such as the non-entailment part of HANS. We show that using our method the increase of OOD accuracy is less correlated with the intrinsic bias as opposed to other methods.
> we have edited the introduction and section 6.1 to a hopefully more clear revision.

---

> > ### Comment · Reviewer_UAbR · 2022-11-22
> > **Thank you for your response**
> >
> > I am still not 100% sure I understand the intuition behind the method, further clarification would be helpful. Moreover, unfortunately given that the revised paper is not available (we are not allowed to consider the github version) and that the new results are not listed in the response, I can't take these comments into consideration.
> >
> > I am therefore keeping my score.

---

> > > ### Author Response · Authors · 2022-12-04
> > > **Clarificaiton**
> > >
> > > Thanks for your response,
> > >
> > > First I would like to note that there is a submitted revised version (but it is results are not fully completed, see https://openreview.net/forum?id=MFD2b2cwr5d&noteId=_0czi_QsGd).
> > >
> > > > I am still not 100% sure I understand the intuition behind the method, further clarification would be helpful.
> > >
> > >
> > > The intuition is that we want the model to learn non-biased representations.
> > > To do that, we constrain it to learn a similar 'relations network' to another model that does not encode the bias features in its representations (supposedly, if we filter out samples that contain bias signals, then the learned representations will not be affected by the shortcuts in the dataset). So 2 samples that are biased will be represented differently under the unbiased model's representations, and we only transfer this knowledge (that these two samples should be different).

---

### Official Review · Reviewer_2WEz · 2022-10-23

**Confidence:** 3
**Clarity, Quality, Novelty And Reproducibility:** The paper is technically solid and cl…
**Correctness:** 3
**Technical Novelty And Significance:** 3
**Empirical Novelty And Significance:** 3
**Recommendation:** 5

**Strength And Weaknesses:**

Strength:

1. The problem is well-motivated and the paper is easy to follow.
2. The method is simple yet effective.
3. The experiments are carried over several different datasets that cover different types of bias, and the results are convincing.

Weakness:
1.  One limitation of SimReg is that exclude highly-confident examples using a biased model (e.g., tinyBERT) might also filter out some "easy" or "unbiased" examples, because neural models often get over-confident.
2. Some related works/baselines are missing. SimReg share many similarity with knowledge distillation (eq 3 is basically a variant of the typical KD loss). So some short discussion about KD would improve the paper and KD loss might be be used as another model variant. Although "intrinsic debiasing"(manipulating representations) methods are novel in the area of spurious correlation mitigation, they have been widely used in debiasing hate speech. Despite the differences in terms of areas, they might also need to be included in the related works.

**Summary Of The Paper:**

This paper proposes SimReg, a method that achieves the goal of debiasing by increasing(decreasing) the similarity between the final model and an unbiased(biased) model. They explore two different ways to impose the regularization: on model representation or on the gradients.
Except for similarity-based regularization, another key component of this paper is the creation of an unbiased dataset, which is critical for regularization's success.
By manipulating internal model representations, SimReg differs from previous methods that downweigh the importance of training samples during training.
Upon extensive evaluation, SimReg is effective in improving models' OOD performance.


**Summary Of The Review:**

The submission proposes a novel method that is effective in mitigating spurious correlation.  But there are some limitations and missing related works that have to be addressed before acceptance.

---

> ### Author Response · Authors · 2022-11-17
> **Response to reviewer 2WEz**
>
> Thank you for your review! We are glad that you liked our motivation and method. We are working on adding more experiments that we hope you will find insightful and improve the paper's quality.
>
> > One limitation of SimReg is that exclude highly-confident examples using a biased model (e.g., tinyBERT) might also filter out some "easy" or "unbiased" examples, because neural models often get over-confident.
>
> please see the first comment.
>
> > So some short discussion about KD would improve the paper and KD loss might be be used as another model variant
>
> Thanks for your comment, indeed there is some similarity between our approach and KD. We have added it to the related work section.
> As for the KD variant, ConfReg can be considered a KD method and it is included in results.
>
> > Although "intrinsic debiasing"(manipulating representations) methods are novel in the area of spurious correlation mitigation, they have been widely used in debiasing hate speech. Despite the differences in terms of areas, they might also need to be included in the related works
>
> Thank you for your note! Indeed the intrinsic debiasing approach is usually used in social biases, we will mention this in the paper.

---

### Official Review · Reviewer_qJJe · 2022-10-23

**Confidence:** 5
**Correctness:** 2
**Technical Novelty And Significance:** 2
**Empirical Novelty And Significance:** 2
**Recommendation:** 3

**Clarity, Quality, Novelty And Reproducibility:**

## Questions and concern
1. An additional concern is that the guidance model is trained on less data and therefore its representations / gradients might not be of good quality, hindering generalization of the base model. This point might be discussed in the paper.
2. Some of the recent de-biasing baselines would be needed in Table 1 (even RWG would be fine)
3. The following references are missing:
- "Increasing Robustness to Spurious Correlations using Forgettable Examples", https://arxiv.org/pdf/1911.03861.pdf
- "Deep Feature Re-weighting", https://arxiv.org/pdf/2204.02937.pdf
- "Correct-n-Contrast: A Contrastive Approach for Improving Robustness to Spurious Correlations", https://arxiv.org/pdf/2203.01517.pdf


**Strength And Weaknesses:**

Strengths:
- *The motivation of the paper is sound*: Recent approaches (such as Deep Feature Reweighting https://arxiv.org/pdf/2204.02937.pdf) show that de-bias models just modify the last output layer, thus re-weighting robust features. It is natural to ask how to make data representations intrinsically less biased.
- *The paper is well-written*: I found the paper easy to read and self-contained, most of the details are present either in the main text or the appendix.

Weaknesses:
- *Lack of enough empirical support*:
The main hypothesis of this paper is that the guidance model representations are less biased and thus can inform the de-biasing methods into learning representations invariant to the bias features. Right now, I don't feel that this hypothesis is substantiated enough by the experiments. I would first expect convincing evidence supporting the fact that the bias is less extractable from the guidance model. An analysis of bias extractability only comes in Section 6.1, Figure 3, but it is only applied to the full regularized model and not to the "guidance" model. I am equally perplexed by Figure 3: the model that *minimizes* the similarity w.r.t the guidance model seem to have representations that are *less* biased than the model that *maximizes* the similarity w.r.t. the guidance model, which seems against the thesis of the paper? I think that most of the paper should be focused on proving that the representations of the final de-biased model are indeed less biased on several datasets: right now this analysis is confined to one dataset (HANS). It would good to include in this study the simple method RWG where the loss is re-weighted by the size of each group.

- *Results are not strong enough / missing baselines*:
Overall, the results obtained by the proposed de-biased model are not significantly better than other baselines. This would not be such an issue if the paper was focused on training de-biased representations without caring so much for improved performance (although it would leave to the authors to justify why having less biased representations would ultimately be useful). For example, if we compare the results of the paper with https://arxiv.org/pdf/1911.03861.pdf, we see lower performance on HANS and PAWS (unknown bias mitigation setting). It would also be suitable to include results for "Correct-n-Contrast: A Contrastive Approach for Improving Robustness to Spurious Correlations": this method also operates on the representation space, by aligning representations between groups, thus it has an inherent similarity with the proposed approach.

- *Manual hyper-parameter search*:
I am a bit uncertain what is the rationale by which the authors chose their hyper-parameters. In Section 3.2, the authors mention "choosing the threshold c_t is performed manually by plotting the confidence of the bias model over the training set". I am not sure what is the rationale for choosing the threshold c_t, and definitely think that an automatic method to do so should be described and analyzed.


**Summary Of The Paper:**

This paper proposes a method to address the spurious correlations / bias problem in the context of NLU. Starting from the observation in Mendelson & Belinkov (2021) that most de-biasing methods consisting in re-weighting data result in actually more biased representations, they explicitly consider to regularize the representations (or the gradients) of the full model to be more similar to a de-biased "guidance" model (trained on small amounts of de-biased data found by either an oracle or by a biased model), in order to train de-biased representations. The method is tested on NLI, Fact checking and paragraph identification.


**Summary Of The Review:**

This paper focuses on an important problem: how to estimate less biased representations rather than just creating de-biased models by re-weighting features in a biased representation. In its current form, I don't feel confident enough to state that the paper advances on its goals. Moreover, performance is on par - or lower - than standard re-weighting approaches. More empirical evidence to support the validity of the method is needed.

---

> ### Author Response · Authors · 2022-11-17
> **Response to reviewer qJJe**
>
> Thank you for your review, we are glad that you liked the motivation!
> We would like to hear your feedback on the new revision.
>
> > I am equally perplexed by Figure 3: the model that minimizes the similarity w.r.t the guidance model seem to have representations that are less biased than the model that maximizes the similarity w.r.t. the guidance model, which seems against the thesis of the paper?
>
> Note that we minimize similarity w.r.t a biased model (that we get from section 3.1), not the unbiased model (the one from section 3.2). There is no guarantee regarding the intrinsic bias of the resulting regularized model when encouraging a model to distance its representations from a biased model ones.
>
> > The main hypothesis of this paper is that the guidance model representations are less biased and thus can inform the de-biasing methods into learning representations invariant to the bias features. Right now, I don't feel that this hypothesis is substantiated enough by the experiments
>
> We have added to the intrinsic bias analysis (Figure 3) both the bias model (f_b) and the unbiased (f_g) guidance model.
> In the new graph we see that f_g has slightly higher compression, and the resulting regularized models had similar results while having better performance.
>
> In addition, we are running a new experiment where we test our hypothesis further (the representations of our models are less biased) and show the importance of debiasing the representations. We retrain the classification layer of the  resulting models on the whole dataset. So far, we have seen that retraining our models leads to better performance on OOD challenge sets compared to models obtained by other methods.
>
> > Results are not strong enough : Overall, the results obtained by the proposed de-biased model are not significantly better than other baselines.
>
> As for the suggested paper (https://arxiv.org/pdf/1911.03861.pdf): We have comparable results on HANS(68.9% vs 68.2%) . On PAWS we report the f1-score of ‘non-duplicate’ label. If we compare the accuracies, our model obtains 42.9%, (44.2% with gradients) where they obtain 47.6%.
> Following your advice, we have added more focus on the representation analysis.
>
> > missing baselines:  ... It would also be suitable to include results for "Correct-n-Contrast"
>
> We have added Forgetables to the results section. As for other “Correct-and-Contrast”, they did not report results on text datasets, and it may not be trivial to re-implement their approach in our setting.
>
> > An additional concern is that the guidance model is trained on less data and therefore its representations / gradients might not be of good quality, hindering generalization of the base model
>
> please see the first comment.

---

### Official Review · Reviewer_73u6 · 2022-10-27

**Confidence:** 4
**Correctness:** 2
**Technical Novelty And Significance:** 2
**Empirical Novelty And Significance:** 2
**Recommendation:** 3

**Clarity, Quality, Novelty And Reproducibility:**

The paper is confusing, the experimental setup is halfway there and additional evaluations are needed. The novelty appears limited and it is unclear what the results tables are reporting. The authors have shared implementation details in Appendix that should help with reproducibility so that is not much of a concern to me right now.

**Strength And Weaknesses:**

- The paper is very confusing. Exposition lacks structure, claims are made without providing evidence to back them up, and there are some hand wavy arguments in this paper. Here are several examples of these issues:
  - "While improving out of distribution (OOD) performance, it was recently observed that the internal representations of the presumably debiased models are actually more, rather than less biased": Biased to what? Bias is a statistical term that's in reference to something specific.
  - The paper relies on having a _biased_ model and an _unbiased_ model, and what those mean doesn't become clear until Section 3.2
  - It is not clear to me how you train a biased model for the unknown biases. You mention you use TinyBert but that's it.
  - You make a point in Section 3.2 that excluding samples on which a _biased_ model is correct and confident would result in an unbiased dataset. That makes the assumptions that (i) instances where your biased model is correct and confident are actually instances which contain confounding which is not always true (see [1], but even other than that, they could just be correct and confident for the right reason), and (ii) your biased model captured all instances that contain confounding. There's nothing stated as to why these assumptions might be valid.
  - "Choosing the threshold $c_t$ is performed manually by plotting the confidence of the bias model over the training set." Yet I can't tell how Figure 3 tells you that $c_t$ has to be 0.8 and not say 0.7 or 0.9 or something.
  - "Appendix A.4) shows that regularizing only $D^B$ results in better OOD performance, supporting our intuition." Did you use same hyperparameters when just regularizing $D^B$ versus when regularizing all examples or different? How were they chosen?
  - "We repeat some of the experiments using DeBERTa to verify our model is not specific to BERT." How did you select which experiments would be repeated and which not?
  - The results tables show means and error terms but its not clear: (i) are these standard error terms or standard deviation?; (ii) what is the source of randomness?; (iii) are differences in performance statistically significant?
- The paper claims that the proposed method leads to better OOD performance. Better performance on one OOD dataset does not imply better OOD performance. This is like testing the model on one example and claiming 100% accuracy if it correctly classifies that example. But the issue with that is that if the second example is incorrectly classified, it's accuracy would suddenly be 50%. Perhaps this model did better on one OOD dataset, to show that it will likely have better OOD performance overall, you need to evaluate on a battery of OOD datasets. A sound claim would be that it leads to better performance on the specific datasets (which is good).
- In related work, a whole line of work on debiasing methods is missing. Some relevant papers below:
  - Victor Veitch, Alexander D'Amour, Steve Yadlowsky, and Jacob Eisenstein. "Counterfactual Invariance to Spurious Correlations: Why and How to Pass Stress Tests." arXiv preprint arXiv:2106.00545 (2021).
  - Zhao Wang and Aron Culotta. "Robustness to spurious correlations in text classification via automatically generated counterfactuals." In Proceedings of the AAAI Conference on Artificial Intelligence, vol. 35, no. 16, pp. 14024-14031. 2021.
  - Divyansh Kaushik, Amrith Setlur, Eduard H. Hovy, and Zachary Chase Lipton. "Explaining the Efficacy of Counterfactually Augmented Data." In International Conference on Learning Representations. 2021.
  - Zhao Wang and Aron Culotta. "Identifying Spurious Correlations for Robust Text Classification." In Findings of the Association for Computational Linguistics: EMNLP 2020, pp. 3431-3440. 2020.
  - Divyansh Kaushik, Eduard Hovy, and Zachary Lipton. "Learning The Difference That Makes A Difference With Counterfactually-Augmented Data." In International Conference on Learning Representations. 2020.

- Errata:
  - incorrect quotation usage in Line 5 of introduction.
  - Third to last line of introduction: "This is different from previous methods" ---> "This is different from some previous methods" (refer to the citations above that do indeed talk about this)
  - Section 3.1: "In the case of decreaseing" ---> "In the case of decreasing"
  - Section 4.1.1, second paragraph: "As OOD test sets, we use HANS" ---> "As OOD test set, we use HANS"
  - Section 4.1.3: Thorne et al. should be \citep and not \citet
  - Section 4.1.4, second last line: "bias.For" ---> "bias. For"
  - Section 5.2, second last line: "support out hypothesis" ---> "support our hypothesis"

[1] Chiyuan Zhang, Samy Bengio, Moritz Hardt, Benjamin Recht, and Oriol Vinyals. "Understanding deep learning (still) requires rethinking generalization." Communications of the ACM 64, no. 3 (2021): 107-115.

**Summary Of The Paper:**

The paper proposes a method for debiasing internal model components, based on similarity-regularization. This is meant to encourage the internal representations to be either similar to representations of an _unbiased_ model, or dissimilar from representations of a _biased_ model. The authors apply this regularization to either the model representations or its gradients. The paper evaluates the approach on three tasks---natural language inference, fact checking, and paraphrase identification---and demonstrates that it improves performance on one out-of-distribution dataset for each task while incurring little degradation in in-distribution performance.

**Summary Of The Review:**

The paper is very confusing and lacks structure. Statements are often made without providing evidence to back them up. The paper relies on several assumptions but there is nothing stated as to why these assumptions might be valid. It is not clear what the results table shows (mean and standard deviation? mean and standard error?) but it's not clear what the source of randomness is (mean over what, different samples of training data or different seeds of a model or both?) or if the differences in performance are statistically significant. The paper claims that the proposed method leads to better OOD performance, but better performance on one OOD dataset does not imply better OOD performance overall. In related work, a whole line of work on debiasing methods is missing.

---

> ### Author Response · Authors · 2022-11-17
> **Response to reviewer 73u6**
>
> Thanks for your review and the points that you have mentioned. We hope that by the edits to the paper it will be considered less hand-wavy.
>
> > Biased to what? Bias is a statistical term that's in reference to something specific.
>
> Here we talk about spurious correlations in datasets. It is common in the literature to refer to them as “bias” in the dataset. We have revised the introduction and provided an explanation of these terms, hopefully it is more clear now.
>
> > It is not clear to me how you train a biased model for the unknown biases. You mention you use TinyBert but that's it.
>
> We train TinyBERT on the dataset. It was previously shown that limited capacity models such as TinyBERT tend to recover previously known spurious correlations in NLI datasets such as MNLI (and use these correlation in its decision making) [1]. We have revised the paragraph hopefully to a more clear version.
>
> [1] - https://openreview.net/pdf?id=Hf3qXoiNkR
>
> > That makes the assumptions that (i) instances where your biased model is correct and confident are actually instances which contain confounding which is not always true (see [1], but even other than that, they could just be correct and confident for the right reason), and (ii) your biased model captured all instances that contain confounding. There's nothing stated as to why these assumptions might be valid.
>
> You are right, there is no guarantee that we capture all the samples that align with the bias, and there is no guarantee that all the samples that we capture are biased. However, the results indicate that the samples we captured are sufficient for improving performance in OOD sets, and indeed we might have better performance if we could identify each sample correctly.
>
> > "Choosing the threshold  is performed manually by plotting the confidence of the bias model over the training set." Yet I can't tell how Figure 3 tells you that  has to be 0.8 and not say 0.7 or 0.9 or something.
>
> please see the first comment.
>
> > "We repeat some of the experiments using DeBERTa to verify our model is not specific to BERT." How did you select which experiments would be repeated and which not?
>
> The consideration was purely due to time limitation, we will run on the other configurations and report it.
>
> >  Did you use same hyperparameters when just regularizing $\mathcal{D^B}$  versus when regularizing all examples or different? How were they chosen?
>
> We have not performed an extensive hyper-parameter search regularizing all examples. However, the difference is large, so such a search is not expected to change the general conclusion. We will add results with a search and update when they are available.
>
> > The results tables show means and error terms but its not clear: (i) are these standard error terms or standard deviation?; (ii) what is the source of randomness?
>
> The error terms are standard deviation. The source of randomness is different random seeds for training the main model (f_m), that is, different initialization of the model and different training batches. The deviation is measured across 3 different random seeds. We have clarified this, and added this detail to the appendix (Training parameters section) .
>
> > The paper claims that the proposed method leads to better OOD performance. Better performance on one OOD dataset does not imply better OOD performance ... A sound claim would be that it leads to better performance on the specific datasets (which is good).
>
> Thanks for your comment.
> You are right, we simply followed the standard paradigm in the literature. We cannot make a strong case for OOD in general and we have revised our claim accordingly.
>
> In addition, we would like to thank you for the data augmentation line of work, we have mentioned it in the paper. And thanks for the corrections!

---

### Author Response · Authors · 2022-11-17
**General comment**

We thank the reviewers for their insightful feedback.
We have provided an initial response and will continue to work and update the paper.
Based on your feedback, we have added a new experiment where we test the representations of the resulting models further. We test the amount of bias restored when retraining the debiased models. We have re-trained the classification layer of the models on the whole dataset, and see that the representations of the model obtained by SimReg have higher performance on the OOD set when retraining, compared to the other approaches. This lends support to our hypothesis that representation-level debiasing removes bias better than other approaches. We are working on extending this to the other datasets and biases.

### response to some concerns that were shared among reviewers:

manual hyper-parameter search :

We simply seeked a threshold that would select the highly biased examples. We agree that there is some arbitrariness to this choice, but would like to highlight that the choice is based on the training set confidence, without looking at the OOD set. It is extremely difficult to perform model selection for OOD generalization settings without considering an OOD set for selection (see Teney et al.), but we made an effort to do so by only looking at the training set confidence plot.

> why would making a model imitate a weak model (i.e., one trained on less data) result in a stronger model?

Note that we are only imitating the guidance model on the identified biased subset DB, and we are not using first order-similarity, so we are not forcing the model to encode using the same representations as the model with less data trained, we are simply transferring the learned relations between the samples. We see empirically that this approach works and increases the performance on OOD challenge sets  in addition to obtaining better representations.

---

### Decision · Program_Chairs · 2023-01-20

**Decision:**

Reject

**Justification For Why Not Higher Score:**

While the motivation of the paper is good, the paper in its current form doesn't really provide tangible contributions (claims are not well supported by experiments).

**Justification For Why Not Lower Score:**

N/A

**Metareview: Summary, Strengths And Weaknesses:**

This paper addresses the problem of bias (spurious correlation) in the context of natural language understanding. Given the counterintuitive observations made in (Mendelson & Belinkov, 2021) that pushing language models towards debiased regimes actually yields inner reprensentations that are more biased, the authors propose to explicitly regularize the representations of their models to be more similar to a "guidance" model known to be debiased. While the motivation of this work seems sound, the reviewers had various concerns about the execution and the lack of support for some of the claims:
* The authors claimed better OOD performance while evaluating on only one dataset. They should have instead evaluated on a battery of OOD datasets to make such a claim. While the authors later rectified their claim, this also weakens the paper as claiming improved OOD performance was the main empirical claim of the abstract.
* Some of the analyzes of the paper seem to be contradictory to its motivation: "the model that minimizes the similarity w.r.t the guidance model seem to have representations that are less biased than the model that maximizes the similarity w.r.t. the guidance model, which seems against the thesis of the paper?" (Review qJJe). This reviewer wasn't satisfied by the authors' "there is no guarantee" response, as it just sidestepped the reviewer's concern. In a private discussion among reviewers, one reviewer felt the authors often limited themselves to acknowledge identified weaknesses while providing analyses on less important points.
* The reviewers also felt results were not strong enough and lacked important baselines: The authors added more experiments, but this further highlighted that results were weak (e.g., 42.9% accuracy vs. 47.6% in prior work).

While the authors addresses some of the minor points raised the reviewers (e.g., on manual hyper-parameter search), most of the main concerns still remain, and I therefore recommend rejection.